# Unbiased mosaic variant assessment in sperm: a cohort study to test predictability of transmission

**Martin W Breuss**[1,2,3]*[†], **Xiaoxu Yang**[1,2][†], **Valentina Stanley**[1,2], **Jennifer McEvoy-Venneri**[1,2], **Xin Xu**[1,2], **Arlene J Morales**[4], **Joseph G Gleeson**[1,2]*

[1]Rady Children's Institute for Genomic Medicine, San Diego, United States; [2]Department of Neurosciences, University of California, San Diego, La Jolla, United States; [3]Department of Pediatrics, Section of Clinical Genetics and Metabolism, University of Colorado School of Medicine, Aurora, United States; [4]Fertility Specialists Medical Group, San Diego, United States

*For correspondence:
martin.breuss@cuanschutz.edu
(MWB);
jogleeson@health.ucsd.edu
(JGG)

[†]These authors contributed
equally to this work

Reviewing Editor: David
Ginsburg, University of
Michigan–Ann Arbor, United
States

## Abstract

**Background:** De novo mutations underlie individually rare but collectively common pediatric congenital disorders. Some of these mutations can also be detected in tissues and from cells in a parent, where their abundance and tissue distribution can be measured. We previously reported that a subset of these mutations is detectable in sperm from the father, predicted to impact the health of offspring.

**Methods:** As a cohort study, in three independent couples undergoing in vitro fertilization, we first identified male gonadal mosaicism through deep whole genome sequencing. We then confirmed variants and assessed their transmission to preimplantation blastocysts (32 total) through targeted ultra-deep genotyping.

**Results:** Across 55 gonadal mosaic variants, 15 were transmitted to blastocysts for a total of 19 transmission events. This represented an overall predictable but slight undertransmission based upon the measured mutational abundance in sperm. We replicated this conclusion in an independent, previously published family-based cohort.

**Conclusions:** Unbiased preimplantation genetic testing for gonadal mosaicism may represent a feasible approach to reduce the transmission of potentially harmful de novo mutations. This—in turn—could help to reduce their impact on miscarriages and pediatric disease.

**Funding:**
No external funding was received for this work.

## Editor's evaluation

This manuscript confirms and extends a recent study from this same group analyzing mosaicism in sperm and transmission of new mutations to relevant offspring. The current work extends previous analysis of mosaicism in sperm to human blastocysts from in vitro fertilization for three subjects (a total of 55 blastocysts), demonstrating transmission of mosaic mutations at close to expected frequencies. The experiments and analysis are carefully done and of high quality, with potential translational relevance to the diagnosis and prevention of genetic disease by pre-implantation genetic testing for a limited subset of mutations.

## Introduction

Genomic mosaic mutations—present in some but not all cells within a tissue—record the history of embryonic development, environmental exposure, and have a wide range of implications for human health (*Biesecker and Spinner, 2013*; *Paquola et al., 2017*). Mosaic mutations are commonly recognized in cancers or localized overgrowth disorders, such as Proteus, CLOVES, and hemimegalencephaly syndromes (*Lee et al., 2012*; *Lindhurst et al., 2011*; *Goncalves et al., 2018*). Increasingly recognized in more complex diseases such as autism spectrum disorder or structural abnormalities (*Jamuar et al., 2014*; *Rodin et al., 2021,*) mosaic mutations are typically restricted to the individual in which they arise unless they appear prior to the embryonic separation of somatic and germline lineages or within germ cell progenitors. In these cases, gonadal mosaic mutations (comprising gonad specific and gonosomal) have the potential to transmit to offspring. These will appear as a considerable portion of de novo mutations and may result in miscarriage or a congenital or complex disease—often without phenotypes in the parents (*Breuss et al., 2021*; *Yang et al., 2020*).

We and others previously demonstrated that 5–20% of identified pathogenic de novo mutations in a child are detectable in parental tissues, with ejaculated sperm demonstrating the highest rate of occurrence (*Yang et al., 2020*; *Breuss et al., 2020*; *Myers et al., 2018*). Every male harbors up to dozens of such mutations in sperm, which—in contrast to other paternal mutation types that increase with age (*Jónsson et al., 2017*)—contribute a life-long threat of transmission largely independent of paternal age (*Yang et al., 2021*). As such, they are thought to explain, in part, the individually rare but collectively common risk of congenital disorders from de novo mutations (*Deciphering Developmental Disorders, 2017*). Yet, experimental evidence of transmission to a conceptus of in situ identified gonadal mosaic mutations—in contrast to the detection of gonadal mosaicism of already transmitted variants—is lacking. This is a critical point upon which clinical implementation hinges because procedures like preimplantation genetic testing (PGT) could be used to select embryos absent for potentially damaging mutations detected in sperm. Here, we demonstrate that gonadal mosaic mutations detected in sperm from individual males can transmit to their preimplantation blastocysts. We decided to assess this early time point, as it avoids any potential bias introduced by a possible selection of mosaic mutations or their lineages during implantation or survival.

## Methods

### Donor population, recruitment, and sample preparation

For gonadal mosaicism assessment, we recruited three infertile couples (F01, F02, and F03) who had supernumerary blastocysts (minimum of 8) that wanted to donate them to research. The males ('sperm donors') all provided a fresh ejaculate and a somatic sample (blood for F01, saliva for F02 and F03). The age of the three sperm donors was 36, 38, and 50 years for F01, F02, and F03, respectively. In accordance with reported population origin, the three fathers' ancestries were most closely matched to Middle Eastern (F01) or European (F02 and F03) ancestry, employing nearest neighbor analysis of principle components (*Taliun et al., 2017*). One of the three females ('egg donors') was a nonidentified egg donor (F01) and provided blood, another woman was the partner (F02) and provided saliva. The third chose not to provide a somatic sample (F03). DNA was extracted from each parental sample using documented procedures (*Breuss et al., 2020*), and extracted and amplified from vitrified blastocysts by thawing in phosphate-buffered saline supplemented with 5% bovine serum albumin and processing with REPLI-g whole-genome amplification methods (Qiagen, Cat. #150343). Informed consent was obtained from all participants (custodians of the blastocysts) as well as from each participant in a study protocol approved by the University of California, San Diego IRB (140028).

### Whole-genome sequencing and massive parallel sequencing

Whole-genome sequencing (WGS; NovaSeq 6000, Illumina) of the sperm donor samples was performed to approximately 300× (*Supplementary file 1*) as described (*Yang et al., 2021*), then analyzed using the 300× MSMF (MuTect2, Strelka2, MosaicForecast) computational pipeline (*Yang et al., 2021*). This pipeline demonstrates a 90% specificity and sensitivity of >95% for mutations at allelic fractions (AFs) above 0.03 (Ref *Yang et al., 2021*). All putative mosaic mutations (single-nucleotide variants and small insertion–deletion mutations only) as well as 120 common and rare single-nucleotide polymorphisms (SNPs), each detected as heterozygous in only one of the sperm donors, were then subjected to

validation using massive parallel amplicon sequencing (MPAS, AmpliSeq Illumina Custom DNA Panel), to orthogonally assess each mutation in sperm donors, egg donors, and blastocysts. We also included two unrelated controls to control for false-positive calls (*Supplementary file 1*; *Breuss et al., 2022*).

## MPAS data analysis

Mosaic mutations from tissues or blastocysts were confirmed or rejected based on the distribution of reference homozygous mutations and the signal in control samples at the same position as described previously (*Breuss et al., 2022*). The overall number and distribution of mosaic mutations were largely consistent with other unbiased analyses performed previously (*Breuss et al., 2020*; *Yang et al., 2021*). SNPs in nonblastocyst samples were assigned genotypes as reference homozygous, heterozygous, or alternate homozygous, based on the appearance of known genotypes previously determined in WGS data. Blastocysts were determined as genotype negative or positive for the SNPs.

To determine the expected number of transmissions to each blastocyst, the list of gonadal mosaic mutations and their measured AF in sperm for each sperm donor was used to determine the expected probability of transmission. Then, each blastocyst was combined with all other blastocysts from a single sperm donor or across all sperm donors as indicated. A Gaussian 1D model was fitted using astropy's LevMarLSQFitter and Gaussian1D to obtain the mean as well as the standard deviation and 95% confidence interval. Code used for data analysis and generation of all plots can be found on GitHub: https://github.com/shishenyxx/Sperm_transmission_mosaicism; *Yang, 2022*. Blastocysts with less than 10% detectable genomic positions according to the panel were excluded from the downstream analysis.

## Determination of false-negative and -positive rate of transmission to blastocysts

To calibrate our genotyping approach on whole-genome-amplified blastocyst material, we determined the transmission of heterozygous variants from sperm and egg donors (*Figure 2—figure supplement 1*). We found that transmission followed expected genetic patterns; furthermore, homozygous variants from a parent—which should be present across all blastocysts—were transmitted in approximately 94% of those analyzed, suggesting a false-negative rate of ~0.06. Importantly, this false-negative rate also includes potential allelic dropout, which can be problematic for single-cell studies or amplification from biopsies. Conversely, when assessing the presence of gonadal mosaic mutations identified in one sperm donor in the blastocysts of the other two, we only found one such event out of 985 possible, suggesting an MPAS false-positive rate of ~0.001. Thus, this approach provides sensitivity to detect clinically relevant gonadal mosaicism (i.e., mutations with a measurable and potentially actionably abundance) and specificity to assess transmissions of perm mosaic mutations to blastocysts, and it suggests that whole-genome-amplified blastocysts exhibit modest allelic dropout, even though allelic imbalance was frequently observed.

## Determination of mosaicism in MPAS data

For each mosaic mutation or SNP, we determined the estimated AF and its 95% confidence interval based on MPAS. As a baseline for observed noise, we determined the distribution of reference homozygous SNPs and their lower 95% confidence interval (population threshold). Mutations considered as a mosaic in a sperm donor's tissue fulfilled three criteria: (1) read depth for the position was at least 100×; (2) the observed 95% confidence interval of the mutation did not overlap with the population threshold or the upper 95% confidence interval of the control samples; (3) either control's lower 95% confidence interval limit had to be below the population threshold. For a blastocyst, to be considered positive for the mutation, similar criteria were applied, but the lower 95% confidence interval could not overlap with 0.05 AF and the read depth at this position had to be equal or above 20×. For SNPs in blood, saliva, and sperm, a mutation was considered heterozygous if above or equal to 0.2 and below 0.8 AF; it was considered homozygous if above or equal to 0.8 AF. Blastocysts were considered positive (either heterozygous or homozygous, not distinguished) if above 0.05 AF.

## Determination of evenness of transmitted mutations across blastocysts

For each of F01, F02, and F03, the observed number of transmitted mutations was randomly assigned to the different blastocysts in 10,000 permutations. For each permutation, the maximum number of

mutations transmitted to one blastocyst as well as the standard deviation of the number of mutations transmitted across each blastocyst was determined. The obtained distribution across the 10,000 permutations was then compared to the observed value. A permutation p value was calculated based on the tail probability (the number of permuted values larger than or equal to the observed value over the total number of permutations).

## Distribution and impact of gonadal mosaic mutations

The 55 gonadal mosaic mutations were distributed across all chromosomes—except for chromosomes 19, 22, and X/Y—roughly as expected based on chromosomal length. As expected based on prior work, the vast majority was found in intergenic ($n$ = 28) or intronic ($n$ = 24) regions; one mutation each was found in a 5'-UTR, the intron of a noncoding RNA, and an exon. The exonic mutation (F03, AF = 0.063) resulted in a nonsynonymous SNV in RABGAP1 (NM_012197; p.Asp74Gly), a rare, previously reported mutation (allele frequency of 7.96e−6) with no known disease association.

## Reanalysis of the REACH cohort

Reanalysis of the REACH cohort data was performed by combining the list of detected gonadal mosaic mutation (*Breuss et al., 2020*; *Yang et al., 2021*) with previously established variants called from the trio WGS (*Brandler et al., 2018*; *Brandler et al., 2016*). For each gonadal mosaic mutation, we determined the occurrence in a child similar to what was done for blastocysts. Finally, the expected probability of transmission was determined as described above.

## Results

We recruited three couples undergoing in vitro fertilization (IVF) for infertility (F01, F02, and F03), where at least eight blastocysts each were donated for research (*Supplementary file 1*). DNA was extracted from each sample, including paternal sperm, using standard procedures (*Breuss et al., 2020*), and blastocysts underwent whole-genome amplification. Tissue-specific and -shared mosaicism was determined for each donor (F01–F03) in sperm and one somatic tissue (i.e., blood or saliva) as described (*Yang et al., 2021*; *Figure 1a*). WGS of sperm samples to 300× read depth allowed 'best practice' detection of mosaicism (*Yang et al., 2021*). The computational pipeline demonstrates

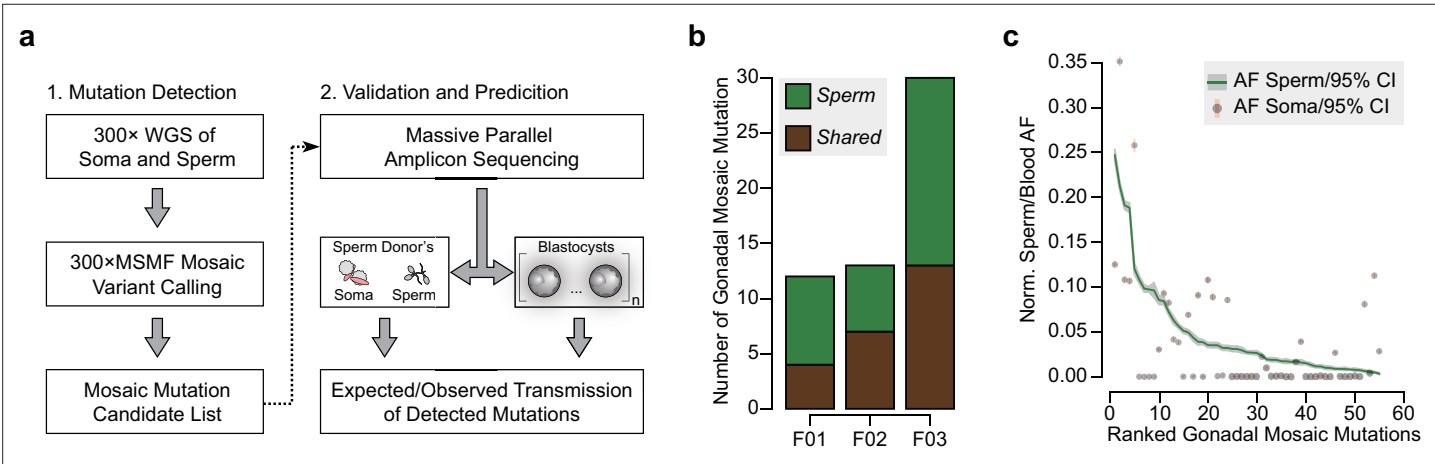

**Figure 1.** Detection of gonadal mosaicism in three sperm donors. (**a**) Overview of the employed workflow from mutation detection to validation and prediction. A single massive parallel amplicon sequencing (MPAS) panel was used for both detections of mutations in parental tissues and in preimplantation blastocysts. WGS: whole-genome sequencing; 300× MSMF: variant calling pipeline on 300× WGS data using MuTect2, Strelka2, and MosaicForecast (*Yang et al., 2021*). (**b**) The number of MPAS-validated gonadal mosaic mutations for each sperm donor, distinguished by color into sperm-specific ('*Sperm*', green) and tissue-shared ('*Shared*', brown) mutations. (**c**) Ranked plot of all gonadal mosaic mutations across the three sperm donors. For each variant, both the allelic fraction (AF; normalized to chromosome count) of the mutation in sperm (green line) and in the soma (brown dot) are shown together with their 95% exact confidence interval. *Shared* mutations tend to be of higher AF compared to *Sperm*.

The online version of this article includes the following figure supplement(s) for figure 1:

**Figure supplement 1.** Detection of mosaic mutations in the three sperm donors.

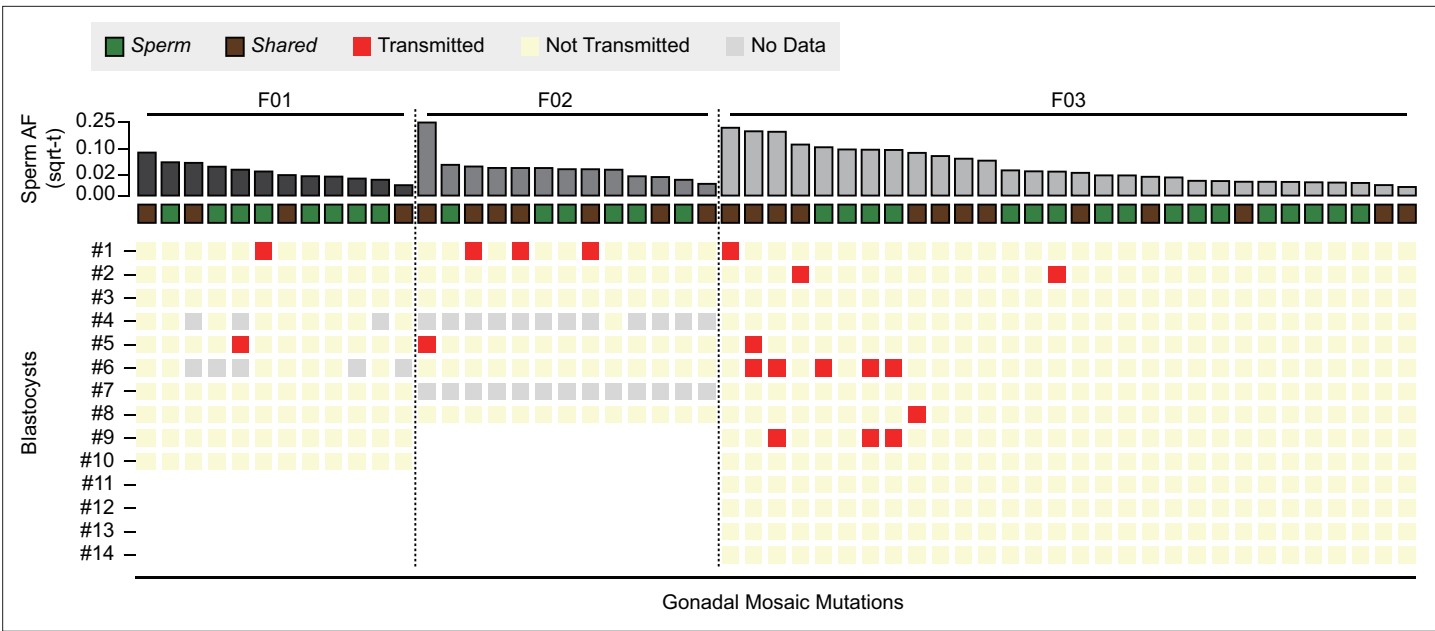

**Figure 2.** Transmission of gonadal mosaic mutations for each sperm donor. Mutations are ranked for each sperm donor by allelic fraction (AF) and visualized as square root-transformed (sqrt-t). Shown are the transmission of 12 mutations across 10 blastocysts (F01), of 13 mutations across 6 blastocysts (F02; 2 of 8 blastocysts did not show sufficient read depth across the mosaic mutations), and of 30 mutations across 14 blastocysts. In total, 19 transmissions of 15 unique mutations were observed. As expected, gonadal mosaic mutations of higher AF are more likely to transmit than those of lower AF. No Data: variant blastocyst pairs, for which read depth was below 20×.

The online version of this article includes the following figure supplement(s) for figure 2:

**Figure supplement 1.** Determination of sensitivity and specificity for mutation detection in blastocysts.

**Figure supplement 2.** Evenness analysis of the distribution of transmitted mutations across blastocysts.

90% specificity and >95% sensitivity for mosaic mutation detection at AFs above 0.03 (*Yang et al., 2021*). All putative mosaic mutations and additional control SNPs from the sperm donors were then subjected to validation using MPAS, AmpliSeq for Illumina Custom DNA Panel (*Breuss et al., 2022*). This was done for both tissues of the sperm donor, one somatic tissue of the egg donor if provided (F01 and F02), and all available blastocysts.

The three sperm donors harbored a combined 55 detected gonadal mosaic mutations—mostly single-nucleotide mutations—with the potential to be transmitted to offspring (F01: 12, F02: 13, and F03: 30) (*Figure 1b*, *Figure 1—figure supplement 1*, and *Supplementary file 1*). While F01 and F02 differed by more than twofold from F03, this likely represents expected biological variability that is within the previously observed range (*Yang et al., 2021*). None of the variants were predicted to impair health when heterozygous, and they, thus, serve as likely neutral variants to model transmission. These mutations were present at AFs between 0.003 and 0.247 (mean: 0.047; standard deviation: 0.055), with those of lower AF typically restricted to sperm ('*Sperm*') and those of higher AF typically found in both sperm and blood or saliva ('*Shared*') (*Figure 1c*, and *Figure 1—figure supplement 1*). Overall, their number, distribution, and abundance were consistent with other unbiased analyses performed previously (*Breuss et al., 2020*; *Yang et al., 2021*). Of note, similar to our prior observations (*Yang et al., 2021*), the sperm donor F01 (36 years of age) had an excessive number of soma-specific mutations ('*Soma*') at lower AFs, evidencing early clonal hematopoiesis (*Breuss et al., 2020*; *Yang et al., 2021*; *Figure 1—figure supplement 1*).

For each sperm donor, 8, 10, or 14 blastocysts, respectively (total: *N* = 32), were assessed for transmission of paternal gonadal mosaic mutations. When calibrating our genotyping approach on whole-genome-amplified blastocyst material, we found a false-negative rate of ~0.06 and a false-positive rate of ~0.001 (*Figure 2—figure supplement 1*). Across all 55 identified gonadal mosaic mutations, we observed 19 transmission events among 15 unique mutations (*Figure 2* and *Supplementary file 1*). For F02, two of the eight blastocysts did not show overall high quality in the MPAS for gonadal mosaic variants and thus were excluded from subsequent analyses.

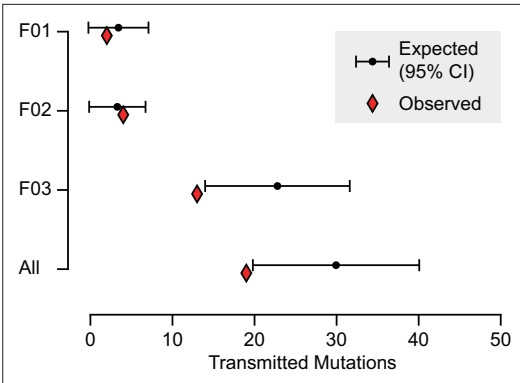

**Figure 3.** Expected and observed transmission of gonadal mosaic mutations to preimplantation blastocysts. The expected number of transmissions—based on the allelic fraction (AF) of all detected mutations in sperm and the number of analyzed blastocysts—is indicated with the mean and a 95% confidence interval for each sperm donor and across all sperm donors. Whereas F01 and F02 transmitted as expected, F03 showed a slight undertransmission, likely related to the nonindependence of mutations due to shared lineage.

The online version of this article includes the following figure supplement(s) for figure 3:

**Figure supplement 1.** Detailed analysis of expected and observed transmission of gonadal mosaic mutations to preimplantation blastocysts.

**Figure supplement 2.** Transmission of gonadal mosaicism in eight previously described families.

Consistent with a model where the risk of transmission is proportional to sperm AF, mutations of higher AF evidenced higher blastocyst transmission rates than those of lower AF ($R = 0.67$, p value = 2.9e−8, Pearson correlation, Methods). Although additional mutations may arise within earlier sperm lineages marked by prior mutations (e.g., blastocyst #2 in F03), somewhat unexpectedly, we observed cosegregation of mutations of almost equal AF (e.g., blastocyst #1 in F02 or blastocyst #6 in F03); this was reflected as a nonrandom distribution of transmitted mutations across blastocysts in F03 (*Figure 2—figure supplement 2*). This suggests that some early male germ cell lineages or individual mitotic divisions are more susceptible to mutation than others, allowing stratification of sperm lineages into those with higher and those with lower mosaic load, and mutations from one sperm progenitor may uncouple during meiosis. Indeed, we observed different combinations of gonadal mosaic mutations across blastocysts (e.g., blastocysts #5, #6, and #9 in F03). This analysis demonstrates that a priori identified gonadal mosaic mutations have the potential to transmit to preimplantation blastocysts.

Based upon their individually measured AFs, we next calculated the expected number of transmission events across all blastocysts. Whereas for both F01 and F02 the observed transmission rate was within the 95% confidence interval of the expectation, for F03 and across all individuals when considered in total, the transmission was slightly below what was expected (*Figure 3* and *Figure 3—figure supplement 1*). This likely reflects a limitation of the model which assumes that gonadal mosaic mutations arise and transmit independently—a simplification, as mutations arising on the same lineage have the potential to cotransmit. This may be especially relevant if sperm lineages do not stochastically accumulate mutations during early development. Nevertheless, our observations closely reflect expectations, suggesting that gonadal mosaicism assessment can serve as a predictor of mutation transmission.

To validate the observation of predictable transmission and slight undertransmission, we further analyzed our previous gonadal mosaicism data from 8 families with a total of 14 offspring, where we reported between 11 and 25 sperm-detectable mosaic variants per father (*Breuss et al., 2020*; *Yang et al., 2021*). We asked whether variants detected in sperm employing our unbiased detection pipeline transmitted to any of the 14 offspring. Across 131 sperm-detectable mosaic variants, we observed nine transmissions among seven unique variants (*Figure 3—figure supplement 2a* and *Supplementary file 1*). As this cohort had a lower number of observable transmission events per paternal sample due to the lower number of fertilization events (1–3 offspring compared to 8–14 blastocysts) and lower sequencing depth, we combined analysis of the eight families. The observed transmission rate was again slightly below the expected 95% confidence interval (*Figure 3—figure supplement 2b, c*). This replication in live-born offspring supports the undertransmission observed in blastocysts and highlights the potential predictive power of gonadal mosaicism assessment.

## Discussion

Here, we directly measured the abundance of gonadal mosaic variants and demonstrate transmission to preimplantation blastocysts for three couples undergoing IVF. These results provide a

proof-of-concept that a priori detected gonadal mosaic mutations can transmit to blastocysts and therefore likely to offspring. There are two important limitations of our study: (1) we were only able to ascertain this phenomenon in three sperm donors and their associated blastocysts; (2) none of the variants interrogated for this study were predicted to be disease causing. Thus, if this approach is to be advanced in the clinic to prevent genetic disease, further research is necessary that expands the size of the cohort and directly assesses a priori identified pathogenic variants for transmission to blastocysts. In addition, further technological development will be required to enable the direct assessment of mutations from biopsies rather than whole blastocysts as implemented in this study.

Previous studies focused on the detection of mosaicism in parents following the identification of de novo mutations in offspring (*Breuss et al., 2020*; *Rahbari et al., 2016*; *Jónsson et al., 2018*). Currently, following the birth of a child with a disease due to a de novo mutation, parents are provided with an empirical recurrence risk of 1–2% for future pregnancies. However, this population risk reflects an average of a near-zero risk in the majority and measurably higher risk (up to 25%) in a minority of instances due to gonadal mosaicism (*Campbell et al., 2014*). While recent studies have improved the risk accuracy by incorporating population models and sampling parental blood (*Jónsson et al., 2018*; *Campbell et al., 2014*), direct assessment of gonadal mosaicism can provide a more precise personalized risk, as well as the potential to prevent recurrence through genetic testing. Yet, this framework is only applicable to families where a de novo mutations has previously been identified, and it can merely prevent recurrence within a family. While we previously demonstrated our ability to detect gonadal mosaicism in males independent of prior transmission (*Breuss et al., 2020*; *Yang et al., 2021*), this study provides the important complement that these detected variants are, indeed, transmitted to the next generation.

What is the potential health impact of the detection of gonadal mosaic variants and their prevention before any child is born with disease? We previously estimated that 1 in 15 males carry a potentially pathogenic mutation, detectable in approximately 5% of sperm cells (*Yang et al., 2021*). Together, this led to a prediction that a gonadal mosaic variant may result in adverse pregnancy or pediatric health outcomes for 1 in 300 concepti (*Yang et al., 2021*). While this likely only represents ~15% of the monogenic sporadic component of diseases such as autism or congenital heart disease, this is the sole fraction that could be prevented with further advances. Thus, if these mosaic mutations were detected prior to pregnancy, and if mutation-carrying blastocysts were identified by PGT, there could be positive consequences for families through the prevention of pregnancy termination or pediatric diseases.

## Acknowledgements

We thank Drs. Yan Ding, Shareef Nahas, and Lucitia van der Kraan (Rady Children's Institute for Genomic Medicine, San Diego) for sequencing and computational support, and Drs. Louise Laurent and Jonathan Sebat (University of California, San Diego) for feedback on the manuscript.

## Additional information

### Competing interests

Martin W Breuss: inventor on a patent (PCT/US2018/024878, WO2018183525A1) filed by UC, San Diego that is titled 'Methods for assessing risk of or diagnosing genetic defects by identifying de novo mutations or somatic mosaic mutations in sperm or somatic tissues. Joseph G Gleeson: Reviewing editor, *eLife*. The other authors declare that no competing interests exist.

### Funding

No external funding was received for this work.

### Author contributions

Martin W Breuss, Conceptualization, Data curation, Formal analysis, Investigation, Supervision, Validation, Writing – original draft, Writing – review and editing; Xiaoxu Yang, Conceptualization, Data curation, Formal analysis, Investigation, Methodology, Software, Validation, Visualization, Writing

– original draft, Writing – review and editing; Valentina Stanley, Data curation, Project administration, Resources; Jennifer McEvoy-Venneri, Data curation, Project administration; Xin Xu, Methodology, Software; Arlene J Morales, Investigation, Project administration, Resources, Supervision, Writing – review and editing; Joseph G Gleeson, Conceptualization, Project administration, Supervision, Writing – original draft, Writing – review and editing

### Author ORCIDs
Martin W Breuss ⓘ http://orcid.org/0000-0003-2200-8604
Xiaoxu Yang ⓘ http://orcid.org/0000-0003-0219-0023
Joseph G Gleeson ⓘ http://orcid.org/0000-0002-0889-9220

### Ethics
Informed consent was obtained from all participants (custodians of the blastocysts) as well as from each participant in a study protocol approved by the University of California, San Diego IRB (140028 ).

### Decision letter and Author response
Decision letter https://doi.org/10.7554/eLife.78459.sa1
Author response https://doi.org/10.7554/eLife.78459.sa2

---

## Additional files

### Supplementary files
• Supplementary file 1. Processed data and sequencing depth information tables. Provided as a separate file; legend contained within.

• Transparent reporting form

### Data availability
Raw whole-genome sequencing data of the bulk sperm, saliva, and blood are available on SRA under accession numbers PRJNA753973 and PRJNA588332. The raw genotyping table is provided as Supplementary File 1. The code for data analysis and statistical tests is available on Github (https://github.com/shishenyxx/Sperm_transmission_mosaicism, copy archived at swh:1:rev:2e10fba00c3c0917d7cf63317f32cc9ed34026d9).

The following dataset was generated:

| Author(s) | Year | Dataset title | Dataset URL | Database and Identifier |
|---|---|---|---|---|
| Breuss MJ, Yang X, Gleeson JG | 2021 | Sperm and soma sample from three males to study transmission mosaicism | https://www.ncbi.nlm.nih.gov/bioproject/PRJNA753973 | NCBI BioProject, PRJNA753973 |

The following previously published dataset was used:

| Author(s) | Year | Dataset title | Dataset URL | Database and Identifier |
|---|---|---|---|---|
| Breuss MJ, Yang X, Gleeson JG | 2019 | Deep whole genome sequencing of sperm and blood samples from fathers of ASD patients | https://www.ncbi.nlm.nih.gov/bioproject/PRJNA588332 | NCBI BioProject, PRJNA588332 |

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
