## [Editor Report]

This manuscript confirms and extends a recent study from this same group analyzing mosaicism in sperm and transmission of new mutations to relevant offspring. The current work extends previous analysis of mosaicism in sperm to human blastocysts from in vitro fertilization for three subjects (a total of 55 blastocysts), demonstrating transmission of mosaic mutations at close to expected frequencies. The experiments and analysis are carefully done and of high quality, with potential translational relevance to the diagnosis and prevention of genetic disease by pre-implantation genetic testing for a limited subset of mutations.

---

## [Decision Letter]

**Decision letter after peer review:**

Thank you for submitting your article "Sperm mosaicism predicts transmission of de novo mutations to human blastocysts" for consideration by *eLife*. Your article has been reviewed by 2 peer reviewers, including David Ginsburg as the Reviewing Editor and Reviewer #1, and the evaluation has been overseen by Ricardo Azziz as the Senior Editor.

Essential Revisions:

1) In Figure 1b – Individual F03 has more than twice as many mosaic variants as F01 or F02. Is there a difference in sequence depth, age, or another explanation? Is the difference statistically significant?

Additional comments/suggestions for authors:

1) The manuscript is well-written and the figures are generally clear. However, this is a specialized field and some of the genetic concepts are quite complex and may be of limited accessibility to the general reader. The authors might consider trying to provide a more general and accessible summary of the work and findings and their implications, including the limitation to only a relatively small subset of de novo genetic mutations.

---

## [Author Response]

Essential revisions:1) In Figure 1b – Individual F03 has more than twice as many mosaic variants as F01 or F02. Is there a difference in sequence depth, age, or another explanation? Is the difference statistically significant?

In our previous work, we found that—on average—a male harbors around 30 mosaic variants in sperm. We did, however, observe a range from 10 to more than 50, and all three individuals would fall within this range (even though F01 and F02 would show numbers at the lower end of the spectrum). We do not believe that this is due to sequencing depth, as F03 is sequenced at a depth between F01 and F02; likewise, we do not believe it to be an age effect: while F03 is older than F01 and F02 (50 years, compared to 36 and 38), the difference is much less than between our young and aged cohorts in our previous publication (~20 years vs. 48+ years). Therefore, we believe these differences to be within the expected biological variation range and not due to a systematic issue. Prompted by this comment we have made two changes to the manuscript:

1) We have now added the ages of the three sperm donors to the Methods section for clarity.

Additional comments/suggestions for authors:1) The manuscript is well-written and the figures are generally clear. However, this is a specialized field and some of the genetic concepts are quite complex and may be of limited accessibility to the general reader. The authors might consider trying to provide a more general and accessible summary of the work and findings and their implications, including the limitation to only a relatively small subset of de novo genetic mutations.